# An interventional study with the Maldives generalist teachers in primary school physical education: An application of self-determination theory

Azeema Abdulla[ORCID]*, Peter R. Whipp, Genevieve McSporran, Timothy Teo

College of Science, Health, Engineering and Education, Murdoch University, Perth, Western Australia, Australia

* Azeema.Abdulla@murdoch.edu.au, azeema70@gmail.com

**Data Availability Statement:** All relevant data are within the manuscript and its Supporting Information files.

## Abstract

In Maldives' primary schools, physical education (PE) is mainly taught by generalist classroom teachers who often lack knowledge and confidence to teach PE. Also, PE programs in primary schools are affected by a perceived lack of infrastructure, resources and equipment. Children in primary schools are allocated one 35 minute period of PE per week. Researchers have previously investigated interventions implemented by specialist PE teachers to enhance the motivation of secondary school students in PE classes. However, limited research has been conducted with generalist teachers' implementing PE intervention with primary school children. In this study we applied self-determination theory to investigate the effects of a professional learning program and an associated resource support package, that was then delivered by the Maldives generalist teachers' delivering PE. The participants were 30 primary school teachers (control group, n = 15; intervention group, n = 15), and their 725 primary school students aged 9–12 years (mean age of 10.5 years). The teachers in the group undertook eight hours of professional learning that focused on strategies and behaviours to support student satisfaction for the three main elements of self-determination theory: autonomy, competence, and relatedness. A repeated measure ANCOVA was carried out for each of the dependent variables. Overall results when compared to pre-intervention measures, the students of teachers in the intervention group significantly increased their post-intervention perceptions for autonomy, competence, and relatedness; and, increased their psychological need satisfaction. Moreover, intervention-students in the post-intervention phase reported reduced need frustration for autonomy, competence, and relatedness; and, experienced higher levels of self-efficacy, enjoyment and engagement. We contend that these results accentuate the usefulness of professional learning programs for generalist teachers delivering PE to promote students' psychological need satisfaction, whilst reducing thwarting behaviours to enhance students' self-determined motivation toward PE classes. The intervention program significantly enhanced the students' perceived need support, and autonomous motivation, it also reduced teachers' need frustrating behaviours within PE classes. Facilitating teachers to provide more moderate to vigorous physical

**Funding:** The author(s) received no specific funding for this work.

**Competing interests:** The authors have declared that no competing interests exist.

activity (MVPA) and psychological need support could reduce the rate of non-communicable diseases that are currently prevalent in the Maldives.

## Introduction

There is a growing body of research worldwide aimed at enhancing secondary students' motivational processes during physical education (PE) classes. Researchers have found that students whose autonomy, competence, and relatedness are satisfied through autonomous motivation are most likely to develop positive adaptive outcomes in PE classes such as enjoyment, engagement, and intention to be physically active in their free time [1, 2]. However, there is limited research on this topic with primary school children taught by generalist classroom teachers. To address the question of what elements of the learning environment enhance primary school students' need satisfaction, need frustration, and self-determined forms of motivation, we deemed it necessary to perform an intervention study with generalist teachers delivering PE in the Maldives. Thus, our aim in this research was to evaluate the effects of a particular intervention that trained and supported generalist teachers of PE in primary schools to promote: their students' autonomy, competence, and relatedness satisfaction; motivational regulation; and adaptive outcomes such as student-perceived self-efficacy, enjoyment and engagement.

### School physical education: The Maldives context

PE is a compulsory subject in Maldives primary schools; however, the schools allocate only one period of 35 minutes per week for PE [3]. The small amount of time allocated to PE presumably reflects a crowded curriculum and a perception that it does not contribute to students' academic achievement. Furthermore, PE is taught by generalist teachers who have had minimal training for the teaching of PE. And, as noted in previous research, generalist teachers teaching PE in primary schools in Male', Maldives, lack the confidence and motivation to teach the subject; typically, a Maldives primary school PE lesson will consist of a short warm-up, followed by teaching of sports skills that focus mainly on ball games [4]. Fitness activities other than running have been observed to be rare in Maldives PE [5]. Maldives generalist teachers often segregate girls, and boys who play ball games such as basketball and handball. Typically, one group of students sat out while the other group played. Most schools had limited playing space and resources [5].

Moreover, schools in the Maldives run as double session schools, with children in grades 6–10 (or 12) attending a morning session and children from grades 1–5 attending an afternoon session. There are no recess or lunch breaks, nor physical activity (PA) between sessions. In Male', the capital city of the Maldives, primary school children averaged only 31.05% of moderate to vigorous physical activity (MVPA) in their short PE classes [5]. Sadly, PE is the only opportunity for most children to be physically active in what is one of the world's most congested cities. Notwithstanding, the children in these challenging circumstances have high motivation, enjoy and look forward to PE classes [6].

### Self-determination theory

To understand the motivational processes of the primary school students in their PE classes, we drew on self-determination theory (SDT). SDT is concerned with the fulfilments of three basic psychological needs (BPNs): autonomy, competence and relatedness satisfaction which

scholars consider to be the building blocks for optimal motivation [7, 8]. In the PE context, autonomy satisfaction refers to when students participate voluntarily in activities and feel like the initiator of their own behaviour; competence satisfaction is embodied in students' feelings of achievement when performing activities; finally, relatedness satisfaction signifies developing feelings of belonging and experiencing positive interactions with classmates.

A sub theory within SDT is cognitive evaluation theory [9] which provides insights into social factors that promote the BPNs outlined in the previous paragraph. For example, the teacher's style of teaching plays a significant role within the PE context. Previous researchers, such as Reeve [10] have described teachers' motivational style as ranging from autonomy-supportive to controlling, thereby either enhancing or hindering students' needs for autonomy, competence and relatedness. Autonomy support refers to teacher actions that take students' viewpoints into account, provide them with freedom of expression, and allow them responsibility for their own decision making, thus promoting the development of autonomous self-regulation [10]. Competence is developed when teachers clearly communicate information on students' performances, enhancing their confidence to achieve the objectives of the lessons. A structured and predictable environment where students are offered assistance during activities and given enough time to achieve goals promotes learning and improvement or competence [11]. Relatedness support refers to when teachers create a learning environment that promotes feelings of connectedness, inclusion, integration, trust, and respect among classmates. Examples of relatedness support include being friendly, close, and empathic towards students and forming groups that facilitate group dynamics [11, 12].

On the other hand, when the teacher uses a controlling motivational style this thwarts the needs of an individual and they will experience need frustration, nonoptimal functioning, and psychological ill-health [13]. Students experience autonomy frustration when teachers enforce disciplinary measures, rely on guilt-induction and criticism of students who fail to meet their expectations [14]. Teachers adopt an autonomy thwarting style through demanding communication, contingent rewards, and requests for students' compliance to the teachers' pre-determined rules [15]. Likewise, teachers with a competence thwarting style ignore students' abilities and individual differences, provide critical and normative feedback, as well as by conducting their classes without any clear structure [16]. Furthermore, relatedness thwarting style teachers are cold and distant from their students [10]. Students in such classes do not feel connected or included in class activities.

Furthermore, SDT envisages individuals as active organisms with inborn tendencies towards development and psychological growth, including different types of motivation that vary according to one's level of self-determined motivation. Explicitly, SDT proposes that behavioural regulation ranges on a continuum from higher (autonomous motivation) to lower (controlled motivation) self-determined forms of motivation, or none at all (amotivation). Autonomous motivation is essentially intrinsic motivation (e.g., students who participate in PE activities for inherent satisfaction, enjoyment and interest) or identified regulation (e.g., students who engage in activities to achieve personally relevant outcomes). Controlled motivation encompasses introjected regulation (e.g., students who try to avoid shame and/or guilt when performing activities) and external regulation or extrinsic motivation (e.g., students who participate in activities to gain external rewards, meet personal expectations and/or avoid punishment). Finally, amotivation is akin to students who are neither intrinsically nor extrinsically motivated as they do not want to participate in an activity at all [7]. Past researchers have found that students who receive BPNs support from their teacher in PE classes have higher forms of self-determined motivation and experience positive adaptive outcomes such as enjoyment, self-efficacy engagement and PA intentions [6, 11, 17–19].

## A review of literature about intervention programs for PE teachers

As mentioned previously, several researchers have assessed the effects of interventions that entail the optimisation of PE teachers' interpersonal style; however, the majority of these intervention studies were conducted with specialised PE teachers at middle and high school levels. For example, Sánchez-Oliva et al. [1] evaluated the effects of an intervention program in which Spanish PE teachers were provided with support strategies in the three BPNs, with the main focus being on the changes in students' motivational process, and adaptive outcomes during PE classes. The results indicated that students of teachers in the intervention group increased their autonomous motivation, controlled motivation, and intention to be physically active much more than students of teachers in the control group. Likewise, Cheon and Reeve [20] and Reeve and Cheon [21] conducted investigations to test the effects of a workshop-based training program that incorporated teaching scenarios, group discussions, and information documents to help PE teachers be more autonomy supportive during their instruction. The findings confirmed that students of teachers in the intervention group increased their levels of autonomous motivation, BPNs satisfaction, future intentions to exercise, and reduced levels of amotivation. Several intervention studies with PE teachers have investigated autonomy-supportive strategies [22], need satisfaction, motivation effects [23, 24] for high school and middle school students. Findings consistently highlighted the utility of teacher training programs as an effective way to enhance students' psychological need satisfaction and self-determined forms of motivation.

Unfortunately, as stated at the outset of this report, there is a paucity of research on interventions concerning the effects of generalist teachers' interpersonal style when teaching PE in primary schools. Among the limited studies, Chang et al. [25] investigated the effect of autonomy support on self-determined motivation in an elementary school in Taiwan. Again, the findings indicated successful development of autonomy support for students in these PE classes, but the researchers did not detail whether the teachers involved were specialised PE teachers or generalist classroom teachers. The study was also limited to autonomy support strategies.

Considering the aforementioned works, our study aimed to make a sizeable contribution to the existing body of knowledge. First, we developed and delivered a professional learning program to improve generalist teachers' interpersonal teaching style, including strategies to promote autonomy, competence, and relatedness support when teaching PE to primary school children. Second, in our study we explored the four levels of SDT [7]: students' support needs (autonomy, competence and relatedness); needs satisfaction and need frustration (autonomy, competence, and relatedness); motivation, (autonomous motivation, controlled motivation and amotivation); and positive adaptive outcomes (self-efficacy, enjoyment and engagement).

In sum, our aim was to investigate the effects of an intervention program in which generalist teachers teaching PE received training to enhance their support strategies for the three BPNs. We evaluated the changes in Maldivian primary school students' motivational processes and adaptive outcomes during their PE lessons. Our hypothesis for the study was that students in the intervention group would: (1) increase their need support and need satisfaction levels, and decrease their need frustration levels; (2) increase their levels of autonomous motivation and decrease scores for controlled motivation and amotivation; (3) attain higher levels of self-efficacy, enjoyment and engagement; (4) and lastly, there would be a significant difference in the between-class variability (in both slope and intercept) within schools.

## Method

Before conducting the research, we obtained ethics approval from the Murdoch University Human Research Ethics Committee and the Ministry of Education of Maldives. The research

involving human data reported in this paper was evaluated and approved by The Murdoch University Human Research Ethics Committee (Approval #: 2018/196). Study protocols were explained to the parents through information letters and opt-out options were given if they did not want their child to participate in the study prior to data collection. The reason for the opt-out option was to give parents a choice about whether they wanted their child to participate or not, in this low-risk study, that may actually have health benefits for their children.

## Participants

**Participant teachers.** All ten public primary schools in the capital city of Male' who use English instructional language were invited to participate in the study. Four schools agreed to participate in the study and these schools had an enrolled population of 800–2000 students. The participant teachers were 30 generalist teachers (29 female; 1 male; mean age 32.33 years; SD = 8.68; range 22–56 years) who taught in the fifth-grade classes of four primary schools in the Maldives capital city of Male'. The teachers had an average of 10 years of teaching experience (SD = 7.06; range = 1–26 years) and held a general primary teaching degree/diploma or a master's degree in education.

**Participant students.** The student sample were all of the students in the fifth-grade 30 classes (n = 725; 377 girls and 348 boys) with a mean age of 10.04 years (SD = 0.39; range = 9–12 years). Each class contained an average of 25 students (range from 22 to 32 students). There were two schools in the control group, with one school providing all seven of its fifth-grade classes for data collection whilst the second control school provided eight classes. The two intervention schools allocated all of their fifth-grade classes to participate in the study. These included five classes and 10 classes from each school, respectively. A robust sample was achieved as 40% of all Male' fifth-grade teachers and students participated in this study.

## Measures for the study

**Perceived support for needs.** Teacher-induced autonomy support was measured using the six-item autonomy support measure proposed by Deci and Ryan [26]. Nine items were proposed by Standage, Duda and Ntoumanis [19] for measuring competence and relatedness support. This instrument comprised the stem: "At the moment, in my PE class. . ." and a response scale anchored at 1 (*strongly disagree*) and 7 (*strongly agree*). Teacher-induced autonomy support (six items) reflected the degree to which students perceived that their PE teacher provided autonomy support behaviours, such as allowing students to make choices, encouraging them to speak out, and listening to their perspectives (e.g., "my PE teacher encourages me to ask questions"). Teacher-induced competence support (four items) indicated whether students perceived that their PE teacher provided clear instruction, guidance, rules, and constructive feedback (e.g., "I felt that my PE teacher liked us to do well"). And, teacher-induced relatedness support (five items) measured the degree to which students perceived that their PE teacher exhibited caring and friendly behaviours (e.g., "I feel that my PE teacher is friendly towards me"). Autonomy, competence, and relatedness subscales were averaged to form a need support composite score. This scale was previously used in the Maldivian context and provided an appropriate Cronbach ($\alpha = 0.979$) [6].

**Basic psychological need satisfaction and frustration.** Students completed the general version of the Basic Psychological Need (BPN) Satisfaction and Frustration Scales [27] questionnaire. Of the 24 items, 12 were about need satisfaction and 12 were about need frustration in one's life, generally. Participants were asked to rate autonomy, competence, and relatedness on a 5-point Likert scale, ranging from 1 (*not at all true*) to 5 (*completely true*). The scores for the three subscales were averaged to form a composite score for both need satisfaction and need frustration.

Questions for need satisfaction included: four autonomy items (e.g., "I feel my decisions reflect what I really want"); four relatedness items (e.g., "I feel connected with people who care for me, and for whom I care"); and four competence items (e.g., I feel capable of what I do"). Questions for the need frustration items included: four autonomy items (e.g., "I feel forced to do many things I wouldn't choose to do"); four relatedness items (e.g., "I have the impression that people I spend time with dislike me"); and four competence items (e.g., "I feel like a failure because of the mistakes I make". For need satisfaction ($\alpha = 0.963$) and need frustration scales ($\alpha = 0.967$), appropriate Cronbach were obtained in the Maldivian context previously [6].

**Motivation.** We used the Children's Perceived Locus of Causality scale (C-PLOC) for PE [28] to measure the different types of motivation. The questionnaire contained 15 items (three items per behavioural regulation) that followed the common stem: "I take part in PE classes. . ."

- intrinsic motivation (e.g., "because I like learning new things")

- identified regulation (e.g., "because it is important for me to do well")

- introjected regulation (e.g., "because I feel guilty when I do not")

- external motivation (e.g., "because I have no choice")

- amotivation (e.g., "I feel I am wasting at time at it"

The students responded to each item on a 4-point Likert scale, ranging from 1 (*strongly disagree*) to 4 (*strongly agree*). The intrinsic and identified regulations were averaged to form an autonomous motivation composite score; and the introjected and external regulation subscales were averaged to form a controlled motivation composite score. Previously in the Maldivian context [6] this instrument showed appropriate Cronbach: autonomous motivation ($\alpha = 0.948$), controlled motivation ($\alpha = 0.938$) and amotivation ($\alpha = 0.924$) respectively.

**Self-efficacy.** To assess students' self-efficacy, we used the questionnaire from Jackson et al. [29] high school PE instrument with a 1 (*no confidence at all*) to 5 (*complete confidence*) Likert scale. In responding to nine items for the self-efficacy construct, the students were instructed to think about their PE class and circle a number that showed how much they believed in their ability in PE lesson at that moment. Items included, "try my hardest in every PE class" and "practice and improve my skills in PE". For this scale, an acceptable Cronbach ($\alpha = 0.942$) had been achieved previously in the Maldivian context [6].

**Enjoyment.** We assessed students' enjoyment of PE with the modified version of the sports enjoyment scale [30]. The four-item scale was modified for the school PE setting, and students were asked to respond to items, such as "my PE lessons are fun to do" and "I think my PE lessons are quite enjoyable" on a seven-point Likert scale anchored at 1 (*not at all true*) to 7 (*very true*). In a previous study, Abdulla et al. [6] obtained an acceptable Cronbach ($\alpha = 0.961$) for the Maldivian context.

**Engagement.** To obtain students' level of in-class behavioural engagement, we asked their teachers to rate the children according to the question: "What level of engagement has each of your students shown in your PE class" on a 7-point Likert scale, anchored at 1 (*no engagement*), 4 (*average engagement*), and 7 (*very high level of engagement*). This behavioural engagement scale had previously been used and shown to be reliable [6, 31].

## Experimental design

The study involved two groups (control and intervention) and two evaluations (pre-test and post -test), perhaps best described as a quasi-experimental, non-equivalent group design. Two

schools were randomly selected for the control group and two schools for the intervention group. The control group comprised 15 teachers and 371 students (female = 189; male = 182; Mage = 10.06, SD = 0.40), and the intervention group 15 teachers and 354 students (female = 187; male = 167; Mage = 10.01, SD = 0.38). The students' pre-test was conducted in early March 2019, the intervention took place from August to October 2019, and the post-test data were collected during the intervention period.

## Intervention program delivered by teachers

The eight-week intervention comprised eight individual lessons designed to increased MVPA in primary school PE taught by generalist teachers. Details of the eight intervention lessons can be found in Abdulla, Whipp and McSporran [32]. Teachers in the intervention group attended two four-hour workshops, both of which were conducted by the lead researcher. The purpose of the workshops was to instruct the intervention group teachers about how to apply the attributes of SDT into their eight intervention lessons designed to increase children's MVPA. Workshops were conducted in a large room, incorporating high-level practical components (i.e. videos, role playing, case studies, group dynamics). These interactive workshop sessions encouraged teachers to ask questions freely.

The main objectives of session one were to explain: the theoretical background used in the study (i.e. the different types of motivational regulations); the influence of PE teacher's interpersonal style (supportive versus controlling style) on BPNs satisfaction and motivational regulation; the importance of BPNs satisfaction to promote self-determined motivation; and the incidence of motivational process on adaptive/maladaptive outcomes in the PE context (confidence, enjoyment, engagement, leisure time physical activity, intention to be physically active).

In the second session, teachers were provided with various strategies to promote students' autonomy, competence, and relatedness satisfaction. Strategies to be autonomy supportive included teachers meeting students' needs, giving students the freedom to make choices, and the benefits of avoiding controlling and pressuring behaviours. Teachers were also guided on strategies for active listening and enhancing students' engagement.

During the design stage of the intervention, to facilitate competence support, a bronze-silver-gold approach to differentiation was used. In the training session, teachers were asked to encourage students to choose achievable gold, silver or bronze challenges/skills thereby giving all students the opportunity to achieve their personal goals. The intervention lessons had clear direction and guidance with achievable short-term goals. Students followed a similar format in all of the lessons, and teachers acted as facilitators. Finally, teachers were given strategies for optimising feedback (before, during and after the task), conducting meaningful assessments, and using adequate communication during the lesson [11].

To promote relatedness support teachers were requested to adopt an empathic attitude such as being close, friendly, and offering help to the students. Additionally, several strategies were offered such as: giving choices to students to form groups, optimising the group's control, and promoting students' social skills (empathy, active listening). In each lesson of the intervention program specific activities were developed to encourage positive group dynamics, trust and corporative skills [1].

## Procedure

The lead researcher explained the study protocols to the students' parents through information letters which outlined the process to opt-out. Goals of the study were explained to the principals and teachers of all ten Male' schools before obtaining consent from four schools to

participate in the study. The teachers and students were assured that the results would be confidential, with no school or individual being identifiable due to all responses being anonymised. The lead researcher administered the surveys, and students had the opportunity to clarify any doubts. The pre- and post-intervention student surveys took approximately 30–35 minutes to complete.

## Data analysis

Data were analysed in two parts: the preliminary analysis, and the intervention effects analysis. As part of the preliminary analysis, we calculated the descriptive statistics of all dependent variables for the pre-test and post-test according to the study groups and for the total sample.

## Preliminary analysis

The descriptive statistics for all of the study variables and internal reliability (Cronbach's alpha) have been presented in Table 1. The Nunnally [33] reliability criterion for all of the self-report measures was 0.70. The validity analyses of the questionnaires were ascertained using AMOS v27 [34]. We used a first-order Confirmatory Factor Analysis, with: a composite of five correlated factors for motivation; three correlated factors for need support, need satisfaction and need frustration; and one factor for self-efficacy and enjoyment. We found that all questionnaires showed a good fit with the data, which exceeded the cut-off [35]. Other aspects considered were: comparative fit index (CFI) > 0.97, Tucker Lewis index (TLI) > 0.97, root mean square error of approximation (RMSEA) < 0.05, and standardised root mean square residual (SRMR) = 0.03. All of the standardised factor loadings of the items were greater than 0.72 and were statistically significant [36]. Then, to determine the possible effect of gender and the dependent variable, we used a one-way Multivariate Analysis of Variance (MANOVA) on the pre-test and post-test separately. MANOVA showed gender was associated with most of the variables at post-test, except for self-efficacy, competence satisfaction, competence frustration,

**Table 1. Descriptive statistics and internal reliability coefficients of the variables in pre-test and post-test.**

| | Total sample (n = 725) | | control (n = 371) | | | | intervention (n = 354) | | | |
| | pre | post | pre | | post | | pre | | post | |
| | M(SD) | M(SD) | M(SD) | α | M(SD) | α | M(SD) | α | M(SD) | α |
|---|---|---|---|---|---|---|---|---|---|---|
| autonomy support | 5.70(1.51) | 5.55(1.35) | 5.71(1.35) | 0.93 | 5.22(1.49) | 0.89 | 5.69(1.66) | 0.96 | 5.89(1.08) | 0.85 |
| competence support | 5.70(1.57) | 6.00(1.25) | 5.70(1.40) | 0.89 | 5.58(1.43) | 0.85 | 5.70(1.73) | 0.95 | 6.44(0.83) | 0.76 |
| relatedness support | 5.71(1.56) | 5.94(1.29) | 5.73(1.42) | 0.93 | 5.61(1.51) | 0.90 | 5.69(1.69) | 0.95 | 6.29(0.90) | 0.83 |
| autonomy need satisfaction | 4.09(0.92) | 4.09(0.92) | 4.10(1.03) | 0.95 | 3.85(0.99) | 0.81 | 4.10(1.09) | 0.91 | 4.35(0.77) | 0.78 |
| competence need satisfaction | 3.99(1.06) | 4.20(0.93) | 4.01(1.04) | 0.91 | 3.98(1.04) | 0.87 | 3.96(1.08) | 0.91 | 4.43(0.73) | 0.82 |
| relatedness need satisfaction | 4.16(1.08) | 4.32(0.92) | 4.19(1.03) | 0.90 | 4.13(1.03) | 0.87 | 4.13(1.13) | 0.94 | 4.53(0.73) | 0.84 |
| autonomy need frustration | 1.87(0.98) | 1.91(1.08) | 1.87(0.87) | 0.87 | 2.17(1.11) | 0.85 | 1.87(1.08) | 0.91 | 1.63(0.98) | 0.95 |
| competence need frustration | 1.82(1.02) | 1.93(0.94) | 1.75(0.92) | 0.89 | 2.13(0.99) | 0.77 | 1.89(1.11) | 0.93 | 1.72(0.83) | 0.77 |
| relatedness need frustration | 1.84(0.99) | 1.76(0.95) | 1.84(0.88) | 0.89 | 1.97(1.03) | 0.78 | 1.84(1.10) | 0.92 | 1.53(0.79) | 0.79 |
| autonomous motivation | 3.36(0.80) | 3.51(0.54) | 3.41(0.73) | 0.93 | 3.41(0.60) | 0.84 | 3.30(0.86) | 0.95 | 3.63(0.44) | 0.78 |
| controlled motivation | 1.59(0.75) | 1.61(0.60) | 1.56(0.67) | 0.89 | 1.69(0.59) | 0.80 | 1.62(0.83) | 0.94 | 1.53(0.60) | 0.80 |
| amotivation | 1.81(0.98) | 1.61(0.69) | 1.76(0.90) | 0.85 | 1.84(0.66) | 0.79 | 1.87(1.06) | 0.94 | 1.37(0.63) | 0.80 |
| self-efficacy | 4.12(0.96) | 4.25(0.75) | 4.17(0.89) | 0.93 | 4.06(0.85) | 0.90 | 4.07(1.03) | 0.94 | 4.46(0.57) | 0.85 |
| enjoyment | 6.04(1.63) | 5.99(1.34) | 6.11(1.55) | 0.96 | 5.60(1.47) | 0.92 | 5.96(1.71) | 0.96 | 6.40(1.06) | 0.88 |
| engagement | 5.54(0.81) | 5.95(0.68) | 5.63(1.46) | - | 5.53(0.55) | - | 5.44(1.67) | - | 6.86(0.35) | - |

Note: Standard deviations are represented in the parentheses.

relatedness frustration and engagement. Gender was included as a covariate in subsequent analyses based on these results.

## Intervention effect analysis

For the intervention effect analysis, we employed a mixed model with repeated measure ANCOVA for each dependent variable, including a between-subjects factor with two covariates (time and gender) (see Table 2). Thus, the data analysis was conducted as a two-level model: repeated measures of each variable, and the growth expected on each subject within the population during the time period under study, were considered as level 1; and between class-variance and the difference in change rate between class in random slope parameters were encompassed as level 2. Each of the analyses had 9 parameters: five fixed effects (intercept, group, time, group*time, and gender) and three random effects (repeated measure variability, between-class intercept variability, and between class slope variability):

- intercept effect was calculated using the estimates of the control group at pre-test (control)

- the group effect was determined by calculating the difference between the intervention group and control group at pre-test

- time effect was estimated using the difference between post-test and pre-test (slope of the control group)

- group*time effect was estimated using the slope difference between the intervention and control group, which indicated the effect of the intervention program.

Under the random effects, within-students variance denotes the students' variability between measures. At the class level, slope variance indicated the between-class variability of

**Table 2. Results of mixed repeated measures ANCOVA in all variables.**

| | | Fixed effect model | | | | | within-students | | between-class | |
|---|---|---|---|---|---|---|---|---|---|---|
| | ICC | intercept | gender | group | time | group*time | effect size | intercept variance | intercept variance | slope variance |
| autonomy support | 0.06 | 5.77** | -0.15* | 0.01 | 0.19* | -0.68** | 0.05 | 0.35** | 0.05 | 0.26** |
| competence support | 0.03 | 5.78** | -0.16* | -0.00 | 0.73** | -0.85** | 0.06 | 0.91** | 0.01 | 0.27** |
| relatedness support | 0.04 | 5.80** | -0.21** | 0.03 | 0.59** | -0.72** | 0.05 | 0.43** | 0.03 | 0.17* |
| autonomy need satisfaction | 0.08 | 4.16** | -0.12* | 0.00 | 0.24** | -0.50** | 0.05 | 0.46** | 0.03 | 0.11* |
| competence need satisfaction | 0.07 | 4.01** | -0.09 | 0.05 | 0.46** | -0.49** | 0.05 | 0.20** | 0.04* | 0.09* |
| relatedness need satisfaction | 0.05 | 4.22** | -0.17** | 0.05 | 0.39** | -0.45** | 0.04 | 0.14** | 0.03* | 0.07* |
| autonomy need frustration | 0.07 | 1.82** | 0.08 | 0.00 | -0.23** | 0.53** | -0.07 | 0.11** | 0.00 | 0.22** |
| competence need frustration | 0.06 | 1.86** | 0.05 | -0.13 | -0.17* | 0.55** | -0.12 | 0.41** | 0.00 | 0.18** |
| relatedness need frustration | 0.05 | 1.80** | 0.08 | 0.00 | -0.31** | 0.44** | -0.11 | 0.44** | 0.00 | 0.14** |
| autonomous motivation | 0.03 | 3.33** | -0.05 | 0.11* | 0.32** | -0.33** | 0.04 | 0.25** | 0.01 | 0.03* |
| controlled motivation | 0.04 | 1.59** | 0.05 | -0.05 | -0.08 | 0.21** | -0.05 | 0.24** | 0.00 | 0.06** |
| amotivation | 0.02 | 1.85** | 0.04 | -0.11 | -0.50** | 0.58** | -0.16 | 0.33** | 0.01 | 0.10** |
| self-efficacy | 0.04 | 4.10** | -0.06 | 0.10 | 0.39** | -0.50** | 0.05 | 0.42** | 0.02* | 0.05* |
| enjoyment | 0.02 | 6.02** | -0.13 | 0.15 | 0.44** | -0.94** | 0.07 | 1.14** | 0.02 | 0.13* |
| engagement | 0.15 | 5.43** | 0.04 | 0.17* | 1.40** | -1.50** | 0.10 | 0.55** | 0.11** | 0.99** |

**p < 0.01

*p < 0.05 gender (0 = female, 1 = male; reference category = female); group (0 = control group, 1 = intervention group; reference category = control group); time (1 = pre-test, 2 = post-test; reference category = pre-test).

the slope, whereas intercept variance indicated the between-class variability of the intercept [37]. Repeated measure variability was analysed with the restricted maximum likelihood estimation (RMLE) method, the autoregressive homogeneous (ARH) covariance structure, and the diagonal covariance type for random effect and Wald test [37].

To calculate the effect sizes via odds ratio, we used the following formula:
$ES = \log(OR) = log\left(a_i d_i \middle/ b_i c_i\right)$, where $a^i$ is the estimation for the control group at pre-test; $d^i$ is the estimation of the intervention group at post-test; $b^i$ is the estimation of the control group at post-test; and $c^i$ is the estimation of intervention group at pre-test. To estimate the magnitude of the effect sizes Cohen criteria for effect sizes (small effect size $< 0.30$; moderate effect size $0.30$–$0.80$; large effect size $> 0.80$) were used [38].

### Intervention effects

In order to determine the possible between-class variations in the intercepts, we performed a series of null models (unconditional intercept only) for each of the variables under investigation through the intraclass correlation coefficient [39]. The results indicated quantification of the degree of between-class variability compared with the variability between students of the same class, thereby indicating the differences in the outcome between the level 2 variables [40]. The intraclass correlation coefficients scores ranged from 0.02 to 0.15 ($Mdn = 0.04$). The intercepts varied significantly across schools in all of the dependent variables ($1.92 <$ Wald Z $< 3.03$; $p < 0.05$), which indicated the development of the multilevel model was warranted [37].

## Results

This study evaluated the impacts of a SDT-informed PE intervention on students in terms of: autonomy, competence, and relatedness; motivational regulation; and adaptive outcomes. The results that follow have been organised under each of these three headings. The results have been presented in the following order. First, for each variable, if there was a significant difference between the intercept for both control and intervention group, this has been identified. Second, the growth rate for each variable for both groups has been provided. Third, random effects intercepts and slope variance for each variable have been provided. Consistent with p-values previously used for intervention studies (1) cut offs of 0.05 and 0.01 were used for significant differences. The effect sizes ranged -0.16 to 0.10.

### Autonomy, competence and relatedness support

For autonomy support there was no significant difference between groups at baseline ($p > 0.05$) (see Fig 1). The intercept for the control group was 5.77 and for the intervention group $5.77 + 0.01$. In terms of growth rate, the control group experienced negative growth (-0.49; $p < 0.01$) whereas students in the intervention group recorded a positive growth (0.19; $p < 0.05$). For competence support, there was no significant difference between the control and intervention group at baseline (intercepts of 5.78 and 5.78 respectively -0.00; $p > 0.05$). The control group grew at a negative -0.12 ($p < 0.01$), whilst the intervention group showed a significantly higher positive score of 0.73 ($p < 0.01$), with a small effect size (ES = 0.06). For relatedness support, prior to intervention the two groups were not significantly different ($p > 0.05$). However, there were significant differences in the growths ($p < 0.01$): the control group had a negative growth of -0.13, whereas the intervention group grew at a rate of 0.59, with a small effect size (ES = 0.05).

As for the random effects, the variability of level 1 was significant in all three need supports ($p < 0.01$). There was no significant effect on the intercept between classes at the beginning of

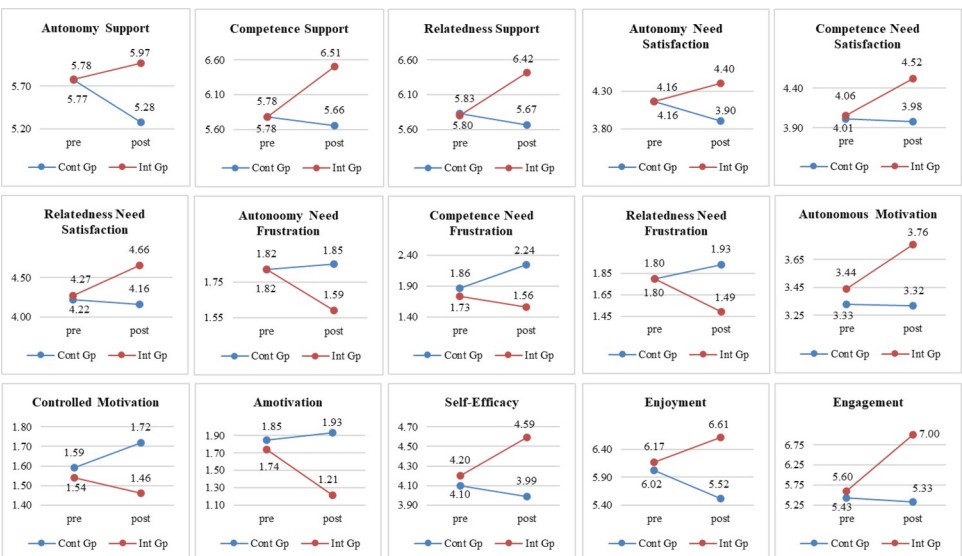

**Fig 1. Scores for control group and intervention group on pre-test and post-test.** (Cont Gp–Control group; Int Gp–Intervention group).

the study. At the same time, the slope varied between classes in autonomy and competence support (p < 0.01) and in relatedness support (p < 0.05).

## Overall need satisfaction relating to autonomy, competence and relatedness

For autonomy satisfaction, as depicted in Fig 1, at baseline the two groups did not differ. The intercept for both the control group and intervention group was 4.16. Group*time was significant (p < 0.01; ES = 0.05): the control group grew at a negative rate of -0.26 while the intervention group grew at the rate of 0.24. For competence satisfaction, the two conditions were similar prior to the intervention (control group = 4.01; intervention group = 4.01+.05 = 4.06). However, there were significant differences in growth rates (p = < 0.01; ES = 0.05): the control group grew at the rate of -0.03 and the intervention group grew at 0.05. For relatedness satisfaction, there was no significant difference between groups before the intervention (p > 0.05). At the end, intercepts for the control group were 4.22, and for the intervention group 4.27 (4.22 + 0.05); thus the control group grew at -0.06 and the intervention group grew at 0.39 (p < 0.01), with a small effect size of 0.04.

In terms of random effects, the variability of level 1 was significant in the three need satisfactions (p < 0.01). At the beginning of the study, the intercept for autonomy satisfaction did not vary, but competence and relatedness satisfaction were significant (p < 0.05) during intervention. Furthermore, the slope varied between classes on all three need satisfactions (p > 0.05).

## Overall need frustration relating to autonomy, competence and relatedness

For autonomy frustration at baseline, the two groups were not significantly different (both control group and intervention group = 1.82; p > 0.05) (see Fig 1). In terms of growth: the control group had a positive growth of 0.03 whereas the intervention had a negative growth (-0.23) with a small effect size (p = < 0.01; -0.07). For competence frustration, at the beginning of the study there were no significant differences (control group = 1.86; intervention

group = 1.73; $p > 0.05$), but the growth rates showed significant differences, with small effect size (ES = -0.12) for the control group (0.38; $p < 0.01$) and intervention group (-0.17; $p < 0.05$). For relatedness frustration, there was no difference at the beginning (intercepts for both groups were 1.80; $p > 0.05$). In contrast, there was a significant difference in growth rates ($p < 0.01$), with a small effect size (ES = -0.11) as the control group grew at the rate of 0.13 and the intervention group grew at the rate of -0.31.

 Regarding the random effects, the variability of level 1 was significant for the three types of need frustrations ($p < 0.01$). The intercept did not vary between classes at the beginning of the study for any of the variables; however, the slope was significantly different for all three variables ($p < 0.01$) between classes.

## Motivational regulation

As shown in Fig 1, for autonomous motivation at baseline the two groups differed significantly (control group = 3.33; intervention group = 3.33 + 0.11 = 3.44; $p < 0.01$). In terms of growth, the control group had a negative growth of -0.01 ($p < 0.01$), and the intervention group grew at the rate of 0.32 ($p < 0.01$), with a small effect size (ES = 0.04). For controlled motivation, at baseline there was no significant difference between the groups (control group = 1.59; intervention group = 1.59–0.05 = 1.54; $p > 0.05$). After the intervention, there was positive growth for the control group (0.13; $p < 0.01$), with no change in growth for the intervention group ($p > 0.05$), with a small effect size (ES = -0.05). For amotivation, at the outset there was no significant difference between the groups (control group = 1.85; intervention group = 1.85–0.11 = 1.74; $p > 0.05$). Afterwards the control group recorded a positive growth rate (0.08; $p < 0.01$) and the intervention group a negative one (-0.50; $p < 0.01$), with a small effect size (ES = -0.16).

 With regard to the random effects, the variability of level 1 was significant for the three motivation types ($p < 0.01$). The intercepts did not vary between classes for any of the motivation variables at baseline; however, slope variance was significant for autonomous motivation ($p < 0.05$), controlled motivation and amotivation ($p < 0.01$).

## Adaptive outcomes

There were no significant differences in the control and intervention groups at baseline for self-efficacy (control group = 4.10; intervention group = 4.10+0.10 = 4.20; $p > 0.05$) (see Fig 1). In terms of growth, the control group's rate was negative (-0.11; $p < 0.01$), while the intervention group grew at 0.39 ($p < 0.01$), with a small effect size (ES = 0.05. Regarding the random effects, the variability of level 1 was significant ($p < 0.01$), with intercept and slope variance significant between classes at the beginning of the study ($p < 0.05$).

 For enjoyment, as depicted in Fig 1, there were no significant differences in intercepts (control group = 6.02; intervention group = 6.17; $p > 0.05$). However, in terms of slope, there were significant differences ($p = 0.01$) in that the control group grew at the rate of -0.94 whilst the intervention group grew at 0.44 and the effect size was small (ES = 0.07). Regarding the random effects, the variability of level 1 was significant ($p < 0.01$), although the intercept did not vary between classes at the beginning of the study ($p > 0.05$), but the slope did vary between classes ($p < 0.05$).

 For engagement, at the beginning of the intervention, the two groups were significantly different ($p < 0.05$) (see Fig 1). Intercepts were 5.43, and 5.60 (5.43–1.17) for the control and intervention groups, respectively. In terms of growth rates, the control group experienced a negative growth (-0.1; $p > 0.01$), while students belonging to the intervention group reported

significantly higher scores after the intervention (1.40; p < 0.01), with a small effect size (ES = 0.10).

## Discussion

Our research entailed two distinct phases. First, we developed and delivered a SDT informed professional learning program for 30 generalist teachers to support their teaching of PE in Maldives primary schools. Second, we evaluated the extent to which the teachers' supportive strategies effected the children's motivational processes and adaptive outcomes in PE classes. We adjusted the baseline scores, which allowed deeper understanding of the effectiveness of the intervention program to enhance the three BPN's satisfaction, student motivation, and adaptive outcomes for students during PE classes. We have reported significant differences and impact of the intervention program when compared to controlled group, we acknowledge the effect size for the data were relatively small, although consistent with a previous intervention study [1].

The post-test scores for the control and intervention groups provided an in-depth understanding of the effectiveness of the intervention program designed for generalist teachers to optimise the BPNs satisfaction, and lower BPNs frustration, whilst targeting student motivation, and adaptive outcomes in PE classes. SDT components were applied throughout the eight-week intervention (one 35 min PE lesson per week for eight weeks) designed to enhance students MVPA. Previously, Abdulla, Whipp and McSporran [32] contended that the intervention which aimed to increase MVPA was highly effective. The percentage of MVPA of the intervention groups increased from an initial pre-intervention of 37.13% to 71.63% post intervention. The generalist teachers from the intervention schools also reported that students showed high levels of motivation and enthusiasm during the PE lessons [32].

The findings supported our first hypothesis, specifically, that the intervention program significantly improved the results for perceived autonomy support and autonomy satisfaction in the students from the intervention group while reducing autonomy need frustration. Furthermore, at the end of the study, students in the control group scored lower in autonomy support and need satisfaction, whilst there was no change to need frustration scores at post-test. After the intervention, students who participated perceived that their teachers supported their self-regulation and were more inclined to listen to their ideas and feelings. In addition, supportive strategies used by the teachers' reflected an increase in students' sense of control and having more autonomy satisfaction. These findings are consistent with several studies undertaken with high school and middle school students where significant improvement to autonomy support and autonomy satisfaction have been reported [1, 20–22, 41]. Furthermore, Cheon, Reeve and Song [42] found an average decrease in students' need frustration (autonomy, competence, relatedness) resulting from a year-long intervention program conducted with PE teachers. Our results also support previous research which reported a correlation between students with improved volition becoming more involved during PE classes and having better attitudes and more enjoyment in extracurricular activities [17].

As for competence support, previous research has identified a lack of significant changes in the perception of competence when implementing short programs that aim to foster BPNs [1, 43]. However, in our study, competence support increased significantly from pre- to post-test for students from the intervention group. The results could be attributed to the following features of the intervention program: clear goals structures and directions, differentiated skill practices, and teachers being free to provide feedback to students. The results indicated that students from the intervention group acquired a higher level of competence support, improved

competence need satisfaction, and reduced competence needs frustration compared those in the control group, thus supporting the prior findings [20, 41, 44, 45].

Finally, the data indicated that our intervention program positively affected relatedness support, related need satisfaction, and reduced relatedness need frustration for students in the intervention group. These positive results may be attributed to the teachers in the intervention program being free to facilitate the lesson instead of being occupied with demonstrating skills and providing instruction. In each lesson of the intervention, a pair/group activity was incorporated, whereupon students were given a choice to select partners they wanted to work with. These strategies allowed students to feel listened to, connected, and wanted in their classes, thus promoting relatedness support, relatedness needs satisfaction and reducing need frustration. Previous researchers obtained similar results with students' increased relatedness satisfaction resulting from specific intervention programs for teachers [41, 46]. The results of our intervention are in line with the postulates of SDT [9], suggesting that the intervention program effectively produced a change in students' perception of autonomy, competence, and relatedness needs, leading to higher BPNs satisfaction and reduced need frustration.

As for motivational regulations, the results supported our second hypothesis. Autonomous motivation increased significantly with the intervention group, whereas for the control group the scores were similar for the pre- and post-condition. Thus, the set of strategies embedded in the intervention program (e.g., differentiated skills, choices, novelty) increased the types of autonomous motivation (intrinsic and identified). In addition, students' feelings of higher autonomy support led to greater autonomy satisfaction and the students developed a greater sense of control during PE classes, thereby enabling them to be intrinsically motivated. These results are similar to those found in previous studies [1, 20, 41, 47] and are consistent with the tenants of self-determination theory [8].

Regarding controlled motivation, the intervention group decreased slightly from pre- to post-data collection, whereas the control group increased from pre- to post-data collection. There was no significant difference at the time (pre and post); however, a significant difference was observed in group*time interaction. During the observation in the controlled group, it was noticed that teachers were assessing students for reporting purposes whereas students in the intervention group were following specific lessons without any assessment. The increase in controlled motivation in the control group could be attributed to where PE teachers grade students publicly, and students must exceed certain thresholds to pass the subject. When teachers in the control group conducted grading, students may have felt externally pressured into action by using internal contingencies which affect their pride and feelings of self-esteem on the one hand, and shame and guilt, on the other [48]. Such factors might have an ascribed increase in controlled motivation in the control group as the examination period approached and during it. The results also fit SDT postulates; although, to date, few researchers have tested the effects of an intervention program with teachers on students' controlled motivation. Tessier, Sarrazin and Ntoumanis [46] claimed that their training program produced a significant decrease in external regulation while not producing change in introjected regulation.

As for amotivation, students in the intervention group decreased their amotivation from pre-to post-test, while students in the control group remained constant. These results are consistent with other studies [20, 41], wherein teachers became more autonomy-supportive and less controlling, and the students' amotivation decreased.

As for third hypothesis, with regard to positive adaptive outcomes, after the intervention, the scores of students from the intervention group had significantly improved for self-efficacy, enjoyment, and engagement. Arguably, these results reflected the benefits attributed to the teachers in the intervention program which enhanced both motivational (BPNs satisfaction and type of motivation) and PE classroom outcomes. When taken together the

aforementioned results demonstrated that when students are intrinsically motivated, they participate in activities with higher enjoyment and engagement [49].

Lastly, regarding the between-class level variability, the results showed significant differences between the two groups in the intercept of competence need satisfaction, relatedness need satisfaction, self-efficacy, and engagement supporting the fourth hypothesis. In contrast, slope variance was significant for all dependent variables. We contend that the approach presented in the intervention program and the valuable role teachers played were central to the intervention's success.

To sum up, the findings confirmed the significance of the teachers' interpersonal styles, and the results are in line with the postulates of SDT. The intervention program effectively produced a change in students' perception of their teachers' ability to promote students' support, need satisfaction, and reduced need frustration on autonomy, competence, relatedness and positive adaptive outcomes for self-efficacy, enjoyment and engagement.

## Strengths and limitations

Among the study's strengths, the intervention program was predesigned to integrate postulates of SDT within the program, and the intervention group teachers were trained on how to apply these components in their classes. The intervention program included strategies to enhance supportive behaviours, and strategies to reduce need thwarting behaviours.

One of the study's limitations was that even though the results showed improvements, they should be interpreted with caution as the effect sizes were small. Another limitation of the study was its multi-component nature as we did not measure the impact of each teaching strategy on the dependent variables. An area for future research would be to design a study with four intervention groups: autonomy-supportive, competence supportive, relatedness supportive, and three-need-supportive in order to compare the effects of each intervention. Lastly, since the teachers in the control group did not receive any support or special attention, the performance of the intervention group teachers may have been partially affected by the Hawthorne effect [50].

## Conclusion

Based on the results of our study, we claim that positive effects resulted from the intervention program with the Maldives teachers, especially for the students' perceived need support, thwarting behaviours, and autonomous motivation within PE classes. Additionally, the results indicated that the PE context could be an interesting means of promoting adaptive outcomes. In this regard, we propose that the Maldives Ministry of Education should consider the possibility of increasing the number of PE periods in their primary schools. They might also consider implementing training for teachers on motivational aspects to improve students' attitudes towards their PE classes, such as self-determined motivation, enjoyment, satisfaction, positive affect, and wellbeing. We assert these actions could enhance both in-school and leisure-time physical activity for students and, ultimately, reduce the rate of non-communicable diseases that are currently prevalent in the Maldives. Lastly, the findings of the present study, confirmed the positive effects of an intervention program with generalist primary teachers on perceived need support while reducing need frustration, autonomous motivation and adaptive outcomes such as self-efficacy, enjoyment and engagement in PE classes thus making a contribution to the existing literature.

## Supporting information

**S1 Dataset.**
(SAV)

## Acknowledgments

We sincerely thank all the teachers and school who took part in the study.

## Author Contributions

**Conceptualization:** Azeema Abdulla.

**Data curation:** Azeema Abdulla.

**Formal analysis:** Azeema Abdulla.

**Investigation:** Azeema Abdulla.

**Methodology:** Azeema Abdulla.

**Project administration:** Azeema Abdulla.

**Resources:** Azeema Abdulla.

**Software:** Azeema Abdulla.

**Supervision:** Peter R. Whipp.

**Validation:** Azeema Abdulla.

**Visualization:** Azeema Abdulla, Timothy Teo.

**Writing – original draft:** Azeema Abdulla.

**Writing – review & editing:** Azeema Abdulla, Peter R. Whipp, Genevieve McSporran.

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
