## [Decision Letter · Decision Letter 0]

17 Feb 2022

PONE-D-22-01130An interventional study with the Maldives generalist teachers in primary school physical education: An application of Self-determination TheoryPLOS ONE

Dear Dr. Abdulla,

Thank you for submitting your manuscript to PLOS ONE. After careful consideration, we feel that it has merit but does not fully meet PLOS ONE’s publication criteria as it currently stands. Therefore, we invite you to submit a revised version of the manuscript that addresses the points raised during the review process.

We look forward to receiving your revised manuscript.

Kind regards,

Ferman Konukman

Academic Editor

PLOS ONE

Journal Requirements:

Reviewers' comments:

Reviewer's Responses to Questions

**Comments to the Author**

1. Is the manuscript technically sound, and do the data support the conclusions?

Reviewer #1: Yes

Reviewer #2: Yes

2. Has the statistical analysis been performed appropriately and rigorously? 

Reviewer #1: Yes

Reviewer #2: Yes

3. Have the authors made all data underlying the findings in their manuscript fully available?

Reviewer #1: Yes

Reviewer #2: Yes

4. Is the manuscript presented in an intelligible fashion and written in standard English?

Reviewer #1: Yes

Reviewer #2: Yes

5. Review Comments to the Author

Reviewer #1: The manuscript is scientifically sound and acceptable

It included all the elements of scientific research

The statistics used are very appropriate

I think the language is good and appropriate

This manuscript is suitable for publication

Reviewer #2: Thank you for your submitted manuscript entitled, “An interventional study with the Maldives generalist teachers in primary school physical education: An application of Self-determination Theory’’. The article is interesting and well written. However, I have a few comments that I suggest you consider before publishing the text.

ABSTRACT

• Clarify the subjects’ level and background

• Why not use average of age, height, body mass and body mass index of participants

• Could be a relevant conclusion of the present study to find what is important to know.

INTRODUCTION

• The introduction is consistent and easy to follow. Hypotheses are clearly formulated.

METHOD

• How was sample size determined? (Sampling technique!)

• Please add the reference number of the ethical approval

• What about the inclusion and exclusion criteria?

STATISTICAL ANALYSIS

• Please, present methods of data analysis and criterion of results interpretation.

• Please add a power analysis, which takes into account the number of variables and the number of cases.

• Please define specificity and sensitivity in this research context

RESULTS

• Obviously, the authors conducted a variance analysis. Please describe and explain the used test(s) in the statistical section.

• Results description is a little chaotic and insufficient. Please, add some introductions to the description of the results and indicate what and why you did. Each result presented in the tables should be commented on in the text. Without that, readers do not know how to interpret the tables. Tables are extensive and maybe it would be valuable to bolded the most important results?

DISCUSSION

• There is insufficient information about effect sizes in the discussion, respectively. Statistically significant results can be not important practically if effect sizes are weak. Please, calculate (if needed) and comment effect size.

• Discussion should be more based on the literature. I suggest considering answers to the following questions when improving discussion: To what extent are the obtained results specific to deliver a training program for 30 generalist teachers to support their teaching of PE in Maldives primary schools.

• The discussion is too long and should be shortened restructured: main results, precise discussion with comparable references, applications for the diagnostic and limitations.

CONCLUSION

• Why might one want to cite this paper? What is the true impact of the literature?

REFERENCES

• Please adapt and improve based on the guidelines of the journal

6. PLOS authors have the option to publish the peer review history of their article (what does this mean?). If published, this will include your full peer review and any attached files.

Reviewer #1: No

Reviewer #2: **Yes: **Souhail Hermassi

---

## [Author Response · Author response to Decision Letter 0]

19 Apr 2022

“An interventional study with the Maldives generalist teachers in primary school physical education: An application of Self-Determination Theory” 

Submitted to PloS One

Comments from the Reviewer Author Response Author Action

(Reviewer 1)

Reviewer #1: The manuscript is scientifically sound and acceptable

It included all the elements of scientific research

The statistics used are very appropriate

I think the language is good and appropriate

This manuscript is suitable for publication 

Reviewer #2: Thank you for your submitted manuscript entitled, “An interventional study with the Maldives generalist teachers in primary school physical education: An application of Self-determination Theory’’. The article is interesting and well written. However, I have a few comments that I suggest you consider before publishing the text. 

Abstract:

Clarify the subjects’ level and background 

Agreed. 

 (Lines 25-29)

In Maldives’ primary schools, physical education (PE) is mainly taught by generalist classroom teachers who often lack knowledge and confidence to teach PE. Also, PE programs in primary schools are affected by a perceived lack of infrastructure, resources and equipment. Children in primary schools are allocated one 35 minute period of PE per week.

Why not use average of age, height, body mass and body mass index of participants We did not collect height, body mass of participants as those variables were not part of the study.

As the reviewer requested mean age is provided.

 (Line 37)

The participants were 30 primary school teachers (control group, n = 15; intervention group, n = 15), and their 725 primary school students aged 9-12 years (mean age of 10.5 years).

Could be a relevant conclusion of the present study to find what is important to know. Agreed

 (Lines 50-55)

The intervention program significantly enhanced the students’ perceived need support, and autonomous motivation, it also reduced teachers’ need frustrating behaviours within PE classes. Facilitating teachers to provide more moderate to vigorous physical activity (MVPA) and psychological need support could reduce the rate of non-communicable diseases that are currently prevalent in the Maldives.

INTRODUCTION

• The introduction is consistent and easy to follow. Hypotheses are clearly formulated. 

METHOD

• How was sample size determined? (Sampling technique!) Agreed

 (Lines 215-218)

Participant teachers. All ten public primary schools in the capital city of Male’ who use English instructional language were invited to participate in the study. Four schools agreed to participate in the study and these schools had an enrolled population of 800-2000 students. 

(Lines 223-231)

Participant students. The student sample were all of the students in the fifth-grade 30 classes (n = 725; 377 girls and 348 boys) with a mean age of 10.04 years (SD = 0.39; range = 9-12 years). Each class contained an average of 25 students (range from 22 to 32 students). There were two schools in the control group, with one school providing all seven of its fifth-grade classes for data collection whilst the second control school provided eight classes. The two intervention schools allocated all of their fifth-grade classes to participate in the study. These included five classes and 10 classes from each school, respectively. A robust sample was achieved as 40% of all Male’ fifth-grade teachers and students participated in this study. 

• Please add the reference number of the ethical approval Agreed

 (Lines 207-209)

The research involving human data reported in this paper was evaluated and approved by The Murdoch University Human Research Ethics Committee (Approval #: 2018/196).

What about the inclusion and exclusion criteria? Answered in the first point of the method section

STATISTICAL ANALYSIS

• Please, present methods of data analysis and criterion of results interpretation. Data were analysed in two parts: the preliminary analysis, and the intervention effects analysis.

(We did not understand very well what the reviewer is mentioning – detail analysis of the data and criterion of results interpretation are provided)

 (Lines 366-384)

Preliminary analysis

The descriptive statistics for all of the study variables and internal reliability (Cronbach’s alpha) have been presented in Table 1. The Nunnally (1978) reliability criterion for all of the self-report measures was 0.70. The validity analyses of the questionnaires were ascertained using AMOS v27 (Arbuckle, 2021). We used a first-order Confirmatory Factor Analysis, with: a composite of five correlated factors for motivation; three correlated factors for need support, need satisfaction and need frustration; and one factor for self-efficacy and enjoyment. We found that all questionnaires showed a good fit with the data, which exceeded the cut-off (Hu & Bentler, 1999). Other aspects considered were: comparative fit index (CFI) > 0.97, Tucker Lewis index (TLI) > 0.97, root mean square error of approximation (RMSEA) < 0.05, and standardised root mean square residual (SRMR) = 0.03. All of the standardised factor loadings of the items were greater than 0.72 and were statistically significant (Hair et al., 2010). Then, to determine the possible effect of gender and the dependent variable, we used a one-way Multivariate Analysis of Variance (MANOVA) on the pre-test and post-test separately. MANOVA showed gender was associated with most of the variables at post-test, except for self-efficacy, competence satisfaction, competence frustration, relatedness frustration and engagement. Gender was included as a covariate in subsequent analyses based on these results.

(Lines 388-420)

Intervention effect analysis

For the intervention effect analysis, we employed a mixed model with repeated measure ANCOVA for each dependent variable, including a between-subjects factor with two covariates (time and gender) (see Table 2). Thus, the data analysis was conducted as a two-level model: repeated measures of each variable, and the growth expected on each subject within the population during the time period under study, were considered as level 1; and between class-variance and the difference in change rate between class in random slope parameters were encompassed as level 2. Each of the analyses had 9 parameters: five fixed effects (intercept, group, time, group*time, and gender) and three random effects (repeated measure variability, between-class intercept variability, and between class slope variability):

 intercept effect was calculated using the estimates of the control group at pre-test (control)

 the group effect was determined by calculating the difference between the intervention group and control group at pre-test

 time effect was estimated using the difference between post-test and pre-test (slope of the control group)

 group*time effect was estimated using the slope difference between the intervention and control group, which indicated the effect of the intervention program. 

Under the random effects, within-students variance denotes the students’ variability between measures. At the class level, slope variance indicated the between-class variability of the slope, whereas intercept variance indicated the between-class variability of the intercept (Heck et al., 2014). Repeated measure variability was analysed with the restricted maximum likelihood estimation (RMLE) method, the autoregressive homogeneous (ARH) covariance structure, and the diagonal covariance type for random effect and Wald test (Heck et al., 2014). 

To calculate the effect sizes via odds ratio, we used the following formula: ES =log⁡〖(OR)=log〗 ├ (aᵢdᵢ⁄bᵢcᵢ┤), where aᵢ is the estimation for the control group at pre-test; dᵢ is the estimation of the intervention group at post-test; bᵢ is the estimation of the control group at post-test; and cᵢ is the estimation of intervention group at pre-test. To estimate the magnitude of the effect sizes Cohen criteria for effect sizes (small effect size < 0.30; moderate effect size 0.30-0.80; large effect size > 0.80) were used (Cohen, 1992). 

Please add a power analysis, which takes into account the number of variables and the number of cases. Conssistent with previous intervention study (Sánchez-Oliva et al., 2017) we have used similar sample size and our sample also produced similar effect size. 

Please define specificity and sensitivity in this research context Agreed

All fifth grade classes from 4 out of 10 schools participated in the study providing 40% of students and teachers teaching those grades, this is robust sample.

Consistent with p-values previously used for intervention studies (Sánchez-Oliva et al., 2017) cut of of 0.05 and 0.01 were used for effect sizes. 

RESULTS

• Obviously, the authors conducted a variance analysis. Please describe and explain the used test(s) in the statistical section. All the statistical tests used are provided with the cut off criteria (Lines 389-398)

For the intervention effect analysis, we employed a mixed model with repeated measure ANCOVA for each dependent variable, including a between-subjects factor with two covariates (time and gender) (see Table 2). Thus, the data analysis was conducted as a two-level model: repeated measures of each variable, and the growth expected on each subject within the population during the time period under study, were considered as level 1; and between class-variance and the difference in change rate between class in random slope parameters were encompassed as level 2. Each of the analyses had 9 parameters: five fixed effects (intercept, group, time, group*time, and gender) and three random effects (repeated measure variability, between-class intercept variability, and between class slope variability):

Results description is a little chaotic and insufficient. Please, add some introductions to the description of the results and indicate what and why you did. Each result presented in the tables should be commented on in the text. Without that, readers do not know how to interpret the tables. Agreed An introduction added to the results section

(Lines 436-445)

This study evaluated the impacts of a SDT-informed PE intervention on students in terms of: autonomy, competence, and relatedness; motivational regulation; and adaptive outcomes. The results that follow have been organised under each of these three headings. The results have been presented in the following order. First, for each variable, if there was a significant difference between the intercept for both control and intervention group, this has been identified. Second, the growth rate for each variable for both groups has been provided. Third, random effects intercepts and slope variance for each variable have been provided. Consistent with p-values previously used for intervention studies (Sánchez-Oliva et al., 2017) cut offs of 0.05 and 0.01 were used for significant differences. The effect sizes ranged -0.16 to 0.10.

Tables are extensive and maybe it would be valuable to bolded the most important results? Most of the results are significant and bold the significant will lead most of the data points bolded. 

DISCUSSION

• There is insufficient information about effect sizes in the discussion, respectively. Statistically significant results can be not important practically if effect sizes are weak. Please, calculate (if needed) and comment effect size. Agreed An introduction line is added

(Lines 541-546)

We adjusted the baseline scores, which allowed deeper understanding of the effectiveness of the intervention program to enhance the three BPN’s satisfaction, student motivation, and adaptive outcomes for students during PE classes. We have reported significant differences and impact of the intervention program when compared to controlled group, we acknowledge the effect size for the data were relatively small, although consistent with a previous intervention study (Sánchez-Oliva et al., 2017)

Discussion should be more based on the literature. I suggest considering answers to the following questions when improving discussion: To what extent are the obtained results specific to deliver a training program for 30 generalist teachers to support their teaching of PE in Maldives primary schools. 

The discussion is too long and should be shortened restructured: main results, precise discussion with comparable references, applications for the diagnostic and limitations. Agreed

The discussion is restructured. 

CONCLUSION

• Why might one want to cite this paper? What is the true impact of the literature? Agreed (Lines 683-687)

Lastly, the findings of the present study, confirmed the positive effects of an intervention program with generalist primary teachers on perceived need support while reducing need frustration, autonomous motivation and adaptive outcomes such as self-efficacy, enjoyment and engagement in PE classes thus making a contribution to the existing literature.

REFERENCES

• Please adapt and improve based on the guidelines of the journal Agreed

In-text and end-text references cited as per the journal requirement

---

## [Editor Report · Decision Letter 1]

22 Apr 2022

An interventional study with the Maldives generalist teachers in primary school physical education: An application of Self-determination Theory

PONE-D-22-01130R1

Dear Dr. Abdulla,

We’re pleased to inform you that your manuscript has been judged scientifically suitable for publication and will be formally accepted for publication once it meets all outstanding technical requirements.

Kind regards,

Ferman Konukman

Academic Editor

PLOS ONE

Additional Editor Comments (optional):

I would like to thank to authors for this interesting and quality study. It is very well designed and manuscript edited professionally. I believe manuscript will provide an important contribution to the literature and field. Best Regards.
---

## [Editor Report · Acceptance letter]

28 Apr 2022

PONE-D-22-01130R1 

An interventional study with the Maldives generalist teachers in primary school physical education: An application of Self-determination Theory 

Dear Dr. Abdulla:

I'm pleased to inform you that your manuscript has been deemed suitable for publication in PLOS ONE. Congratulations! Your manuscript is now with our production department. 

Kind regards, 

on behalf of

Dr. Ferman Konukman 

Academic Editor

PLOS ONE